# Prediction of liver disease in patients whose liver function tests have been checked in primary care: model development and validation using population-based observational cohorts

David J McLernon,[1] Peter T Donnan,[2] Frank M Sullivan,[3] Paul Roderick,[4] William M Rosenberg,[5] Steve D Ryder,[6] John F Dillon[7]

▶ Prepublication history and additional material is available. To view please visit the journal (http://dx.doi.org/10.1136/bmjopen-2014-004837).

For numbered affiliations see end of article.

**Correspondence to**
Dr David J McLernon;
d.mclernon@abdn.ac.uk

## ABSTRACT

**Objective:** To derive and validate a clinical prediction model to estimate the risk of liver disease diagnosis following liver function tests (LFTs) and to convert the model to a simplified scoring tool for use in primary care.

**Design:** Population-based observational cohort study of patients in Tayside Scotland identified as having their LFTs performed in primary care and followed for 2 years. Biochemistry data were linked to secondary care, prescriptions and mortality data to ascertain baseline characteristics of the derivation cohort. A separate validation cohort was obtained from 19 general practices across the rest of Scotland to externally validate the final model.

**Setting:** Primary care, Tayside, Scotland.

**Participants:** Derivation cohort: LFT results from 310 511 patients. After exclusions (including: patients under 16 years, patients having initial LFTs measured in secondary care, bilirubin >35 μmol/L, liver complications within 6 weeks and history of a liver condition), the derivation cohort contained 95 977 patients with no clinically apparent liver condition. Validation cohort: after exclusions, this cohort contained 11 653 patients.

**Primary and secondary outcome measures:** Diagnosis of a liver condition within 2 years.

**Results:** From the derivation cohort (n=95 977), 481 (0.5%) were diagnosed with a liver disease. The model showed good discrimination (C-statistic=0.78). Given the low prevalence of liver disease, the negative predictive values were high. Positive predictive values were low but rose to 20–30% for high-risk patients.

**Conclusions:** This study successfully developed and validated a clinical prediction model and subsequent scoring tool, the Algorithm for Liver Function Investigations (ALFI), which can predict liver disease risk in patients with no clinically obvious liver disease who had their initial LFTs taken in primary care. ALFI can help general practitioners focus referral on a small subset of patients with higher predicted risk while continuing to address modifiable liver disease risk factors in those at lower risk.

### Strengths and limitations of this study

- Our study has successfully developed and validated the first risk prediction model and subsequent user-friendly scoring tool, the Algorithm for Liver Function Investigations, for liver condition diagnosis in patients with no obvious liver condition at the time of incident liver function testing in primary care.
- This model can be used to facilitate general practitioner decision-making about whom to refer to secondary care.
- γ-Glutamyltransferase (GGT) was found to be a powerful predictor of liver disease, over and above alkaline phosphatase and transaminase. Since some laboratories do not routinely test for GGT, its use should be re-evaluated.
- The observational data lacked some potential predictors of liver disease, for example, alcohol intake and body mass index. However, other available predictors such as liver function tests and deprivation may act as surrogate markers for such factors.

## BACKGROUND

General practitioners (GPs) commonly request liver function tests (LFTs), but the results rarely identify cases of liver disease.[1] Abnormal LFTs can be indicators of many different diseases besides those of the liver, such as metastatic malignancy, congestive heart failure and inflammatory or infective conditions.[2–5] It can therefore be difficult for a GP to identify a specific liver disease in patients with abnormal LFTs, particularly in those who do not have pathognomonic signs or symptoms. This leads to variation in approaches to further investigation with some patients being inadequately investigated and others at low risk of disease having invasive tests with attendant morbidity and

cost.[6] The size of this problem is huge with 22% of patients having their initial panel of LFTs performed in general practice with at least one abnormality in Tayside, Scotland.[7] Referral of all these patients to secondary care would place a large burden on healthcare system resources, so guidance is needed to ensure that care is focused on those most in need.

The primary care management of these types of patients might be facilitated through the use of a clinical prediction model which estimates the risk of a specific outcome adjusted for patient characteristics.[8] These models, such as the Framingham cardiovascular disease risk score,[9] and FRAX for osteoporotic fracture, are regularly used in primary care.[10] It is important that clinical prediction models are assessed for their predictive ability, preferably on an external cohort.[11]

This population-based historical cohort study followed up two separate cohorts of patients living in Scotland with no clinically recognised liver disease who initially had LFTs undertaken in primary care.[12] The aims of the study were:

1. To derive a clinical prediction model using a cohort of patients from Tayside that would estimate the risk of a liver disease over the 2 years following LFTs taken in primary care;
2. To validate this model geographically and temporally using a different cohort of patients from 19 general practices in Scotland;
3. To convert this model into a user-friendly clinical scoring tool that will allow the GP to easily calculate the risk of liver disease diagnosis at 6 months and 2 years.

## METHODS
Separate populations were used to develop the prognostic model (derivation cohort) and then validate it (validation cohort).

### Derivation cohort
The study population was initially derived from a laboratory database which contains all electronically available LFT results from patients within Tayside during the 15-year period from January 1989 to December 2003 linked to hospital admission and GP prescribing data.[12] LFTs included bilirubin, albumin, alkaline phosphatase (ALP), γ-glutamyltransferase (GGT), alanine transaminase and aspartate aminotransferase. Since many laboratories only measure either alanine transaminase or aspartate aminotransferase, these two tests were combined as one test and are referred to as transaminases throughout.

Patients aged 16 and above with no obvious or reported clinical signs of a liver disease on presentation to their GP, and with at least two different LFTs requested from the index appointment between 1989 and 2003, were eligible for inclusion. The following exclusion criteria ensured that the study population of patients had no clinically recognised liver disease at presentation in primary care:

▶ Bilirubin greater than 35 μmol/L at baseline, suggesting jaundice.
▶ Diagnosis of ascites, encephalopathy, varices or portal hypertension within 6 weeks of their first LFTs.
▶ History of any liver disease before baseline.

### Databases
The databases relevant to this study are described briefly in box 1 and in further detail elsewhere.[12–18] In Scotland, all individuals registered with a GP have a unique identifier, the Community Health Index (CHI).[14] This number is used for all health encounters

---

**Box 1** Databases record linked to create the Tayside derivation cohort

1. Death registry from the General Register Office for Scotland including date and causes of death.
2. Deprivation score was assessed by the Carstairs method.[13] The score was divided into two categories of affluent and deprived.
3. Tayside Community Health Index (CHI) files: Contains CHI number,[14] name, address and date of birth for all individuals registered with a general practitioner in Tayside. It also holds migration dates to and from Tayside. The database was pseudonymised before release to the researcher.
4. Regional biochemistry database: All liver function tests (LFTs) came from the largest hospital laboratory in the Tayside region, Ninewells. Two smaller hospital laboratories contributed electronic LFT results later, one from 1998 and the other from 2003.
5. Scottish Morbidity Records 1 (SMR01) database: Patient-specific morbidity data are routinely collected in Scotland, and collectively these are known as SMR. SMR01 contains hospital admissions and procedures for all hospitals in Scotland and is one of the oldest and most complete national health datasets in the world.[15] Major comorbidity groups at baseline were identified using this database, and the Diabetes Audit and Research Tayside Scotland (DARTS) and Heart disease Evidence-based Audit and Research Tayside Scotland (HEARTS) databases for diabetes and chronic heart disease, respectively.[16]
6. SMR04 database: Holds all inpatient and day case episodes for mental health specialties, which identified patients admitted/discharged with diagnoses of alcohol dependency or drug misuse.
7. Pharmacist dispensed prescription database: A community-based database that holds encashed prescriptions in Tayside,[17] and allowed identification of patients on some potentially hepatotoxic drugs including statins, antibiotics and non-steroidal anti-inflammatory drugs.
8. Epidemiology of Liver Disease In Tayside (ELDIT) database: Contains all Tayside patients with liver diseases who have been 'electronically' diagnosed using record linkage of biomedical datasets, including virology and immunology.[18] ELDIT was used for the study exclusion criteria (ie, previous liver disease, liver disease complications within 6 weeks) and for the outcome of liver disease diagnosis during the 2-year follow-up period.

and is contained within all of the databases used in this study. Since it contains person identifiable information, the CHI was mapped to a project-specific pseudonymous 10-digit code by the data protection officer at the Health Informatics Centre, University of Dundee, before removing the CHI and releasing the data to the researcher. All of the electronic databases were electronically record-linked deterministically using this project-specific pseudonymous code.

## Ethics statement

Written informed consent from patients was waived by the Tayside Committee for Medical Research Ethics because the databases were anonymised so that no patient identifiable information was accessible. The databases used in this study (see box 1) covered the entire study period and were used in accordance with procedures approved under the Caldicott Guardian and the Data Protection Act UK (1998), in line with the European directive of 1995.

## Baseline characteristics

As well as the five analysed LFTs, baseline characteristics included age, gender, deprivation,[13] comorbidities (including cancer, diabetes, ischaemic heart disease (IHD), stroke, renal disease, respiratory disease and biliary disease), diagnosed alcohol and drug dependency, methadone use, and the use of statins, non-steroidal anti-inflammatory drugs (NSAIDs) or antibiotics in the 3 months before LFTs. Since patients with bilirubin >35 μmol/L were excluded, bilirubin was categorised into normal and mildly raised (<18 and 18–35 μmol/L, respectively, for male patients; <16 and 16–35 μmol/L, respectively, for female patients).

## Outcome

The primary outcome for this study was liver disease diagnosis during the 2 years following the initial analysis of LFTs in primary care. Liver diseases were identified from the Epidemiology of Liver Disease in Tayside (ELDIT) database (box 1), described in further detail elsewhere.[18] A detailed table of the liver diseases included and their source database is presented in online supplementary appendix 1.

## Model derivation

Survival analysis was conducted using parametric regression models to estimate the risk of liver disease within 2 years. The starting point was taken as the date of the initial LFT test and the endpoint for each patient was whichever one of the following events came first: end of follow-up (ie, 2 years later), death, end of study period (ie, 31 December 2003), date of emigration or liver disease diagnosed. All patients whose endpoint was not liver disease diagnosis were censored in the model.

The Weibull regression model was used for model building using potential predictors at baseline. A manual stepwise technique was used to arrive at a model

that contained only significant predictive characteristics. The functional form of continuous characteristics (age, albumin, ALP, GGT and transaminase) was assessed by plotting each against the Martingale residuals and, subsequently, appropriate transformations were carried out where necessary. Clinically important two-way predictor interactions were also investigated and were included in the final model if they were statistically significant. The significant predictors from the Weibull model were then refitted to different parametric model distributions including the generalised γ, log-logistic, log-normal and exponential distributions to find the one that fitted best.[19] The Akaike's information criterion (AIC) was used to select the optimal model. Covariates which were just outside the significance level for the Weibull model were also added to these other models to check whether they became significant. If they did, then they were included in that model.

The problem of missing data occurs in almost all retrospective studies using routine health databases. The simplest way of dealing with this is to use only cases with complete data in the analysis. However, this leads to the loss of potentially valuable information from the incomplete cases (and hence loss of power), and introduces bias, especially if there are systematic differences between the complete and incomplete cases. Therefore, the findings from an analysis using only the complete cases may not be a true reflection on what would be found if all the cases were analysed. To analyse only the complete data assumes that the missing data are missing completely at random (MCAR), which is unlikely.[20 21] A weaker version of the MCAR assumption is the missing at random (MAR) assumption. This differs from MCAR in that it assumes that the missing data are dependent on one or more variables in the observed data.[21] Assuming MAR, a multiple imputation technique using the Markov chain Monte Carlo method was conducted to impute missing values for the LFTs using PROC MI in SAS.[22] Every model was fitted to 30 imputed datasets and the 30 sets of parameter estimates and covariances were combined to produce inferential results using PROC MIANALYZE. All baseline characteristics, time to liver disease (or censored event) and liver disease diagnosis outcome were included in the procedure. The complete data were also analysed separately as a sensitivity analysis.

The integrated discrimination index (IDI) was used to measure the improvement in the final model for each individual covariate.[23] The IDI for a covariate is essentially the difference between the proportion of variance explained by the full model (ie, adjusted for all covariates) and the model without the covariate of interest. The sensitivity, specificity, positive predictive value (PPV) and negative predictive value (NPV) were also calculated for different risk cut-offs, accounting for censoring.[23]

## Model validation

A separate cohort of patients was obtained from the Primary Care Clinical Informatics Unit (PCCIU),

University of Aberdeen,[24] and was used to externally validate the final model. The validation cohort contained all patients registered with 19 practices from across Scotland excluding Tayside. The practices were participating in the Practice Team Information project operated by the Information Services Division of the National Health Service National Services Scotland, and contributed data to PCCIU. The patient population within the PCCIU database is broadly representative of the Scottish population with respect to age, sex and social deprivation.[25] The validation cohort contained patients having their initial LFTs measured in primary care between January 2004 and August 2008. All eligible patients had to have test results for ALP, bilirubin, albumin and transaminase. All baseline characteristics and outcome data obtained for the derivation cohort were also obtained for the validation cohort. The same exclusion criteria listed above were also applied to the validation cohort.

The parameter estimates of the final model from the derivation cohort were applied to the validation cohort to assess its predictive ability. The C-statistic was used as a measure of discrimination.[26] Discrimination assesses the model's ability to correctly distinguish between patients who develop liver disease (for whom the model assigns a high risk) and do not develop liver disease (for whom the model assigns a low risk). The model's predicted probabilities were assessed for accuracy using calibration plots.[27] A calibration slope test was conducted to test for overfitting of the model and true differences in the effects of predictors.[27] This was done by fitting the linear predictor of the final model in a model by itself and testing its slope. A significant deviation from one signifies overfitting. The sensitivity, specificity, PPV and NPV of the model were also calculated for the validation cohort. These assessments were calculated using the validation cohort with the average GGT values across the 30 imputed datasets used for those patients who did not have a GGT measurement.

### Decision curve analysis

Decision curve analysis was used to determine a range of threshold predicted probabilities of liver disease where the primary care decision to refer a patient to secondary care would be better than assuming all patients are disease free (ie, not referring anyone) and assuming that all patients have liver disease (ie, referring everyone).[28] The method involves plotting the net benefit of the model against the threshold probability. The net benefit is defined as the difference between the proportion of patients who are true positive and false positive weighted by the relative harm of a false-positive and false-negative result. The net benefit of the model is then compared with the scenario where everyone is assumed to be disease free and therefore not referred (net benefit equals zero) and to the scenario where everyone is assumed to be disease positive and therefore referred. Threshold probabilities that have a higher net benefit than both of these scenarios means that, for these

thresholds, the model is better than referring no one and referring everyone. The analysis was conducted using the validation cohort and the predicted probabilities at the maximum follow-up time of 2 years were used.

### Clinical scoring tool

A point-based scoring tool was created from a simpler version of the final model for potential use in primary care.[29] Continuous predictors from the model were dichotomised into clinically and statistically sensible categories to create the tool. Mid-points of the categories were used to estimate the risk of liver disease for each category. The tool estimated short-term risk at 6 months follow-up and longer term risk at 2 years follow-up.

Model development and validation were performed using SAS (V.9.3) (SAS Institute, Cary, North Carolina, USA) and the decision curve analysis was performed using the stdca() function in R V.3.0.1.[30–32]

## RESULTS

### Baseline characteristics

Before applying the exclusion criteria, LFTs were extracted for 310 511 patients. After excluding patients under 16 years of age, non-Tayside residents and those whose initial LFTs were measured in secondary care, 99 165 patients remained. After excluding those with clinically recognised liver disease at baseline, the derivation cohort contained 95 977 patients with incident initial LFTs taken in primary care and with no obvious liver disease. There were more female patients (57.9%) than male patients (42.1%), and the median (IQR) age was 54.6 (39.2–68.8) years (table 1). The most frequent comorbidity was IHD (5.6%), followed by cancer (3.8%).

Only 8388 (8.7%) patients had all five LFTs. The percentage of complete data for each LFT was as follows: ALP (99.2%), albumin (99.2%), bilirubin (93.6%), transaminases (76.5%) and GGT (10.9%). There were more male patients with complete data (ie, having all five LFTs) than female patients (54.6% vs 45.4%) compared with the incomplete data group (40.9% vs 59.1%). The group with complete data was also more deprived and contained more alcohol-dependent patients than the incomplete data group (see online supplementary appendix 2). When those without GGT measurements were compared with those with GGT measurements, the results were similar to the above since those without GGT measurements comprised the majority of the incomplete data group. The group of patients without transaminase measurements contained a higher proportion of female patients (63.5% vs 36.5%) than the group with transaminase measurements (56.2% vs 43.8%) and had a higher median ALP result (86 vs 73 U/L).

### Liver disease diagnosis

A total of 481 patients (0.5%) were diagnosed with a liver disease during the 2-year follow-up period. Of these, 339 (70.5%) had at least one abnormal LFT (out

**Table 1** Baseline characteristics of patients in the derivation (n=95 977) and validation (n=11 653) cohorts

| Baseline characteristics | Cohort n (%) or median (IQR) | |
| --- | --- | --- |
| | Derivation | Validation |
| Age (years) | 54.6 (39.2–68.8) | 60.0 (47.0, 72.0) |
| Gender | | |
| Male | 40 374 (42.1) | 5271 (45.2) |
| Female | 55 603 (57.9) | 6382 (54.8) |
| Carstairs category* | | |
| Affluent | 47 286 (49.3) | 2753 (23.6) |
| Deprived | 48 691 (50.7) | 8900 (76.4) |
| Comorbidity history | | |
| Cancer† | 3629 (3.8) | 956 (8.2) |
| Diabetes | 1386 (1.4) | 1441 (12.4) |
| IHD | 5370 (5.6) | 2034 (17.5) |
| Renal disease | 141 (0.2) | 155 (1.3) |
| Respiratory disease | 2636 (2.8) | 883 (7.6) |
| Stroke | 1471 (1.5) | 583 (5.0) |
| Medication in previous 3 months | | |
| Statins | 3176 (3.3) | 3178 (27.3) |
| NSAIDs | 6698 (7.0) | 1762 (15.1) |
| Antibiotics | 8307 (8.7) | 1962 (16.8) |
| Abusive substance | | |
| Alcohol | 2632 (2.7) | 465 (4.0) |
| Drug | 371 (0.4) | 0 (0.0) |
| Methadone | 377 (0.4) | 10 (0.1) |
| Liver function tests | | |
| Albumin (g/L) | 44.0 (42.0–46.0) | 44.0 (41.0, 46.0) |
| ALP (U/L) | 76.0 (62.0–94.0) | 75.0 (62.0, 92.0) |
| Transaminase (U/L) | 18.0 (14.0–26.0) | 21.0 (16.0, 30.0) |
| GGT (U/L) | 26.0 (17.0–47.0) | 27.0 (18.0, 45.0) |
| Bilirubin‡ | | |
| Normal | 81 111 (91.0) | 10 587 (90.8) |
| Mildly raised | 8058 (9.0) | 1066 (9.2) |

Data reported are median (IQR) or percentage.
*Carstairs categories 1–3 were recoded as affluent and categories 4–7 were recoded as deprived.
†Not including biliary cancer or hepatocellular cancer.
‡Normal bilirubin defined as 0–15 μmol/L for female patients, 0–17 μmol/L for male patients; mildly raised bilirubin defined as 16–35 μmol/L for female patients, 18–35 μmol/L for male patients.
ALP, alkaline phosphatase; GGT, γ-glutamyltransferase; IHD, ischaemic heart disease; NSAID, non-steroidal anti-inflammatory drug.

of 22 673). Cirrhosis was the most frequently diagnosed condition (n=75, 15.6%), and in 19 of these patients the cause was recorded as alcohol-related liver disease. Primary biliary cirrhosis was diagnosed in 73 patients but only as definite in 13 (probable n=37; possible n=23). The next most frequent diagnoses were hepatitis C (n=68), alcoholic-related liver disease (n=68), hepatitis B (n=40), hepatocellular carcinoma (n=34), autoimmune hepatitis (n=29) and fatty liver disease (n=28).

## Prediction of a liver disease

The log-normal regression model had the lowest AIC and therefore was proven to have the best fit to the survival distribution. The following baseline characteristics were significant predictors of liver disease diagnosis: increasing GGT, decreasing albumin, alcohol dependency, being female, increasing ALP, living in a deprived area and younger age (table 2). There were also significant interactions between albumin and transaminase, and albumin and methadone use. NSAID use, antibiotic use, drug dependency and all comorbidity history indicators were not significantly predictive of a liver disease. The IDI statistic showed that albumin explained the greatest percentage of variance in the model followed by GGT. When the model was fitted to the subgroup of patients with complete data (N=8738), only GGT and albumin were significant out of all the predictors. However, the parameter estimates had the same direction of effect and a reasonably similar size of effect. The added heterogeneity from the subgroup of patients with imputed GGT and the increased power from the much larger study population would explain the significance of all the predictors in the imputed model.

## Validation

After exclusions, the external cohort contained 11 653 patients with incident initial LFTs taken in primary care. A total of 57 patients (0.5%) were diagnosed with a liver

**Table 2** Parameter estimates (95% CI) and IDI for the final log-normal regression model predicting risk of a liver disease diagnosis within 2 years of initial liver function tests (481 diagnosed)

| Parameter | Coefficient (95% CI) | p Value | IDI (%) |
|---|---|---|---|
| Intercept | 9.524 (1.986 to 17.062) | 0.013 | |
| Albumin | 0.488 (0.306 to 0.669) | <0.001 | 0.711* |
| Log(GGT) | −1.704 (−2.223 to −1.184) | <0.001 | 0.689 |
| Methadone (yes vs no) | 8.319 (−2.019 to 18.657) | 0.115 | 0.465* |
| Log(ALP) | −0.739 (−1.213 to −0.264) | 0.002 | 0.307 |
| Log(transaminase) | 2.016 (−0.151 to 4.183) | 0.068 | 0.266* |
| Alcohol dependent (yes vs no) | −1.210 (−1.856 to −0.563) | <0.001 | 0.143 |
| Gender (male vs female) | 0.583 (0.236 to 0.930) | 0.001 | 0.135 |
| Age at baseline | 0.014 (0.004 to 0.024) | 0.009 | 0.121 |
| Deprived† (yes vs no) | −0.518 (−0.852 to −0.183) | 0.002 | 0.013 |
| Methadone×albumin | −0.295 (−0.530 to −0.061) | 0.014 | |
| Albumin×log (transaminase) | −0.070 (−0.122 to −0.018) | 0.008 | |
| Scale | 4.551 (4.192 to 4.910) | <0.001 | |

Parameter estimates are in decreasing order of IDI. A negative coefficient indicates an increased risk of diagnosis, while a positive coefficient indicates a decreased risk of diagnosis. However, this differs for terms involved in interactions, that is, increasing transaminase increases risk; for a methadone user, increasing albumin increases risk; for non-methadone users, decreasing albumin increases risk.
*The relative interaction terms containing this parameter were also excluded.
†Carstairs categories 1–3 were coded as affluent and categories 4–7 were coded as deprived.
ALP, alkaline phosphatase; GGT, γ-glutamyltransferase; IDI, integrated discrimination index.

disease within 2 years. The proportion of male and female patients was reasonably similar to the derivation cohort (45.2% vs 42.1% male patients), but the external cohort was older, more deprived and had a greater proportion of comorbidities and medications (table 1). However, the median LFTs were similar between the two cohorts. GGT was missing for 4178 (35.9%) patients and was imputed using the same method as for the derivation cohort. The average GGT measure for each patient from the 30 imputed datasets was used where the original measure was missing so that the validation could be performed in one dataset.

The C-statistic for the final model parameter estimates applied to the external cohort was 0.78 (95% CI 0.72 to 0.84). The calibration plot (see online supplementary appendix 3) showed some visible evidence of overfitting for the highest risk group; however, the calibration slope showed no significant deviation from one (slope=0.871 (95% CI 0.696 to 1.047); p=0.15).

The sensitivity, specificity, PPV and NPV for different cut-offs of predicted risk of liver disease diagnosis, along with the receiver operating characteristic curves for the final model applied to the derivation and external validation cohorts, are displayed in online supplementary appendix 4. A cut-off greater than or equal to the 75th centile of the risk score (ie, 0.43% in the derivation cohort and 0.57% in the validation cohort) had sensitivity and specificity of approximately 75% each. The NPV was very high at almost 100% for all cut-offs. Although PPV was very low, it rose to over 10% for predicted risks over 5% and reached 20–30% for higher risks (over 20%). These performance measures were similar for derivation and validation cohorts. From the decision curve in figure 1, it can be seen that the range of threshold predicted probabilities where the prediction model is of value is approximately between 0.5% and 7.5%.

### Predicted probabilities

The final model and how to calculate the risk of liver disease directly from it is presented in online supplementary appendix 5. Figure 2 shows the risk of diagnosis over 2 years for the average risk patient and two example patients with different characteristics as described in the legend.

### Clinical scoring tool

A user-friendly paper-based scoring tool was developed from a slightly simpler version of the final model with the albumin and transaminase interaction term removed

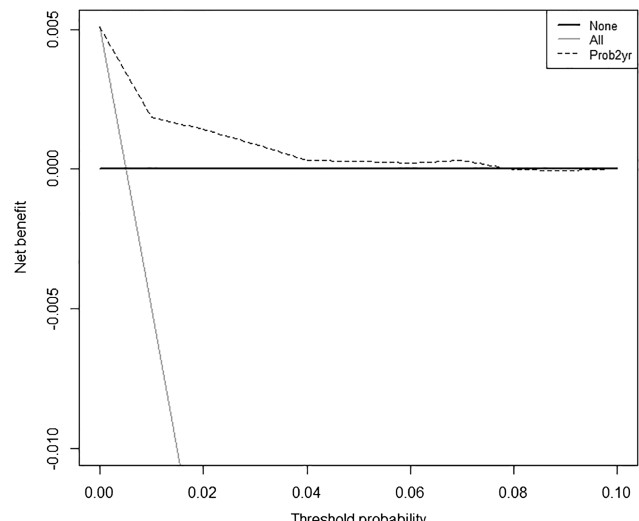

**Figure 1** Decision curve for a model to predict liver disease diagnosis in patients having their liver function tests (LFTs) measured in primary care. Dashed line: prediction model; grey line: assume all patients have liver disease; black line: assume no patients have liver disease.

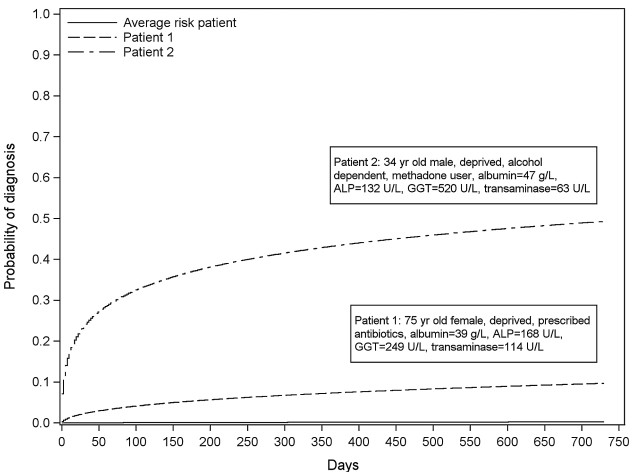

**Figure 2** Probability of a liver disease diagnosis during the 2 years for the average risk patient and two example patients with different risk levels. Note: Average risk is defined as the risk calculated using the average linear predictor value from the model. Patient 1 and 2 characteristics are presented in the boxes within the plot. The estimated risk of being diagnosed with a liver disease within 2 years for each of these patients is 0.002 (average risk patient), 0.097 (patient 1) and 0.493 (patient 2). ALP, alkaline phosphatase; GGT, γ-glutamyltransferase.

(figure 3). The C-statistic for the scoring tool when applied to the validation cohort was 0.77 (95% CI 0.59 to 0.89). When the tool's approximation to the linear predictor was used in a calibration slope test, it was not significantly different from one, β=0.93 (0.73 to 1.14), p=0.53. The scores for each of the risk factors add up to a total score that corresponds to the risk of liver disease. The baseline case for the tool was a 45-year-old woman, living in an affluent area, who was not alcohol or methadone dependent with transaminase=11.5 U/L, ALP=46 U/L, GGT=17 U/L and albumin=48 g/L. Her score of zero corresponded to a liver disease risk of 0.03%. As an example, a 60-year-old man living in a deprived area, who is alcohol dependent but not a methadone user, with transaminase=70 U/L, ALP=130 U/L, GGT=45 U/L and albumin=38, has a score of 29, which corresponds to a liver disease risk within 2 years of 4.1%. For comparison, when this patient's characteristics were entered into the full model in online supplementary appendix 5, their risk was 3.8%.

The correlation between the predicted probabilities from the scoring tool probabilities and the predicted probabilities from the final model when applied to the validation cohort was 0.91. Online supplementary appendix 6 shows the scatterplot of the relationship between the two.

## DISCUSSION
### Summary
A user-friendly clinical scoring tool has been developed that can estimate the risk of liver disease diagnosis at

6 months and 2 years in patients who had their initial LFTs measured in primary care. The tool is based on a clinical prediction model which was derived and validated using large population databases. This model adds morbidity and a longer duration to a previous algorithm that predicts 1-year mortality as part of our Algorithm for Liver Function Investigations (ALFI).[33]

### Strengths and limitations
This is the first large-scale population-based analysis of LFTs with complete determination of liver disease outcome. ALFI has been derived from unselected 'real-world' observations in a geographically defined population: an approach recommended by the National Institutes of Health.[34] A weakness of the electronic databases was the lack of some potential predictors of liver disease, for example, alcohol intake and body mass index, which the GP can readily ascertain to guide decisions. However, predictors in the model, such as LFTs and deprivation, may act as surrogate markers for such factors. While we have no data on the clinical indications for LFTs being measured, we were able to identify patients with a history of major comorbidities since 1980, including cancer, diabetes and IHD using hospital admission records and population registers, and patients who were prescribed statins, NSAIDs and antibiotics. Using the predictors we did have, the model was proven to have good ability to predict liver disease. There will have been an underascertainment of some liver diseases; particularly those diagnosed in primary care, such as mild alcohol-related liver disease and the steatotic end of the spectrum of non-alcoholic fatty liver disease. Patients with these diseases are most likely to be dealt with by being advised to lose weight or reduce alcohol intake, and only those who developed further problems and who were referred would be captured on the database.

The aim of the study was to combine known significant predictors of liver disease into one source of prognostic information that could provide the GP with an individualised estimate of the predicted risk of liver disease at the point where the patient has had their initial LFTs measured. If, at that point, the patient has a high risk of liver disease based on this model, then the GP can also use other contextual information and their own clinical judgement to make a decision as to whether to monitor or refer the patient to a liver clinic according to national or local protocols.

The predicted probabilities from the scoring tool correlated well with those from the model. However, it is evident from the scatterplot in online supplementary appendix 6 that for high predicted probabilities from the final model the tool underestimates the predicted probability. This is because many of these higher probabilities come from patients who have very highly abnormal LFTs. The tool assigns such patients the points based on the highest cut-off in the tool. For example, if a patient has an ALP measure of 825, the tool assigns the same points as someone with an ALP of 181 (ie, +4

| Risk factors | | Categories and corresponding score | | | | | | Score |
|---|---|---|---|---|---|---|---|---|
| Age (Years) | 16 to 35 | 36 to 55 | 56 to 75 | ≥75 | | | | |
| | +1 | 0 | -1 | -2 | | | | |
| Transaminase (U/L) | <20 | 20 to 39 | 40 to 59 | 60 to 79 | ≥80 | | | |
| | 0 | +3 | +5 | +6 | +7 | | | |
| ALP (U/L) | <60 | 60 to 89 | 90 to 119 | 120 to 149 | 150 to 179 | ≥180 | | |
| | 0 | +1 | +2 | +3 | +3 | +4 | | |
| GGT (U/L) | <30 | 30 to 59 | 60 to 89 | 90 to 119 | 120 to 149 | ≥150 | | |
| | 0 | +6 | +9 | +12 | +13 | +14 | | |
| Albumin (g/L)  AND | <31 | 31 to 35 | 36 to 40 | 41 to 45 | ≥46 | | | |
| Methadone user | +19 | +20 | +21 | +21 | +22 | | | |
| OR | | | | | | | | Or |
| Not methadone user | +19 | +15 | +10 | +5 | 0 | | | |
| Gender | Male | Female | | | | | | |
| | -2 | 0 | | | | | | |
| Deprived | No | Yes | | | | | | |
| | 0 | +2 | | | | | | |
| Alcohol dependent | No | Yes | | | | | | |
| | 0 | +5 | | | | | | |
| | | | | | | | Total score | |

**Estimated probability of liver disease diagnosis at 6 months and 2 years**

| Total Score | Liver disease probability (%) | | Total Score | Liver disease probability (%) | |
|---|---|---|---|---|---|
| | within 6 months | within 2 years | | within 6 months | within 2 years |
| 5 or less | <0.1 | <0.1 | 26 | 1.3 | 2.7 |
| 6 | <0.1 | 0.1 | 27 | 1.5 | 3.1 |
| 7 | <0.1 | 0.1 | 28 | 1.8 | 3.6 |
| 8 | 0.1 | 0.1 | 29 | 2.0 | 4.1 |
| 9 | 0.1 | 0.2 | 30 | 2.3 | 4.6 |
| 10 | 0.1 | 0.2 | 31 | 2.7 | 5.2 |
| 11 | 0.1 | 0.3 | 32 | 3.1 | 5.9 |
| 12 | 0.1 | 0.3 | 33 | 3.5 | 6.6 |
| 13 | 0.1 | 0.4 | 34 | 4.0 | 7.4 |
| 14 | 0.2 | 0.4 | 35 | 4.5 | 8.2 |
| 15 | 0.2 | 0.5 | 36 | 5.1 | 9.2 |
| 16 | 0.2 | 0.6 | 37 | 5.7 | 10.2 |
| 17 | 0.3 | 0.7 | 38 | 6.4 | 11.2 |
| 18 | 0.4 | 0.8 | 39 | 7.2 | 12.4 |
| 19 | 0.4 | 1.0 | 40 | 8.0 | 13.6 |
| 20 | 0.5 | 1.2 | 41 | 9.0 | 15.0 |
| 21 | 0.6 | 1.3 | 42 | 10.0 | 16.4 |
| 22 | 0.7 | 1.6 | 43 | 11.0 | 17.9 |
| 23 | 0.8 | 1.8 | 44 | 12.2 | 19.5 |
| 24 | 1.0 | 2.1 | 45 or more | >13.3 | >21.0 |
| 25 | 1.1 | 2.4 | | | |

**Figure 3** Clinical scoring tool for likely liver disease diagnosis in primary care. For each risk factor category, enter the corresponding score into the box on the right-hand side. Sum the scores in the total score box. Look for the total score in the lower table and read off the risk of liver disease within 6 months and/or 2 years. ALP, alkaline phosphatase; GGT, γ-glutamyltransferase.

points) since ≥180 U/L is the highest cut-off in the tool. It would be inefficient to expand the number of groups for the LFTs to include such large values as the score card would be huge. In such circumstances, the GP would automatically refer without the need for a scoring tool. Also, it is advisable to reduce the influence of extreme values when creating such a scoring tool by using the 99th centile value of measurements as the largest possible value when creating the cut-offs.[29]

### External validation
The validation cohort came from 19 general practices across nine different regions of Scotland during a different time period than the derivation cohort. Good

discrimination and calibration demonstrated that ALFI is transportable to different geographical and temporal populations. The C-statistic for discriminatory ability was 0.78, which is comparable to that found in the validation of the Framingham equation on different cohorts ranging from 0.63 to 0.83.[35] The external cohort also contained more deprived and comorbid patients than the derivation cohort, proving that ALFI is also robust to changes in baseline characteristics.

### GGT and transaminase
GGT and, to a lesser extent, transaminase were either missing or not tested for a large proportion of patients in the derivation cohort, but much less so in the

validation cohort. From the database, it was not possible to determine the exact reason for the non-presence of these two LFTs. However, in Tayside, the laboratories do not routinely include GGT with the other four LFT results unless specifically requested by the primary care physician. The demographics of the patients with complete data (ie, males, illicit drug users, alcohol dependants and patients living in deprived areas) suggested that some primary care physicians may have requested GGT where they suspected substance abuse.[7] Testing bone biochemistry may explain the reason for transaminase not being measured, since this group contained a higher proportion of female patients who are more susceptible to bone disease, such as osteoporosis, and had a higher median ALP, which is a marker for bone disease. Furthermore, one of the hospitals in Tayside analysed transaminase using a separate analyser for several years throughout the study and did not keep electronic copies, which may also explain some of the missing values. Therefore it was assumed that the missing/untested LFT data depended on variables in the observed data, the assumption required for multiple imputation,[22] and that the appropriate guidelines for handling this problem were followed.[36]

### Implications for practice and research

At a cut-off of 0.6% (the 75th centile) for liver disease risk, sensitivity and specificity were similar with values of 73.7% and 75.3% in the validation cohort. However, the PPV for low cut-offs such as this was poor. The specificity at a cut-off of 1.2% risk is greater than 90% and rises to 100% at the 10% cut-off. The NPV of the model is very high even for reasonably small cut-offs of risk, meaning that the model is good at ruling out risk of liver disease within 2 years. However, this is not surprising, given the low prevalence of liver disease in the sample of 0.5%.[37] The PPV was low since liver disease diagnosis was relatively rare, but it rose to over 10% for predicted risks of over 5% and in a small minority of patients with higher predicted risk reached 20–30% with the NPV remaining very high. Therefore, ALFI may be most accurate at higher predicted risk cut-offs with reasonable PPVs. As well as displaying reasonable accuracy, it is also important to show that the model has the potential to improve decision-making.

The decision curve analysis showed that the model was of value for threshold probabilities between 0.5% and 7.5%. To determine whether the model is of clinical value, we should consider the possible range of threshold probabilities of liver disease at which GPs would decide to refer the patient to secondary care. The decision to refer is not a particularly risky intervention for the patient. Initially, it will lead to further blood tests and possible ultrasound in secondary care. The outcome of these results will determine whether the patient requires a liver biopsy which has some risk attached to it. However, if the patient gets to that stage, then the clinician must have a high suspicion of liver disease, meaning that the decision to refer was correct. If the GP does not refer and the patient does have or develop liver disease, then the risk to the patient (ie, of being a false negative) depends on the type and stage of the disease. Therefore, the GP may think that the intervention of referral is much less risky than not referring and decide that a probability threshold of, say, 5% is enough to refer. For thresholds between 5% and 7.5%, the model is also reasonably accurate. Therefore, ALFI could help GPs focus referral on a small subset of patients with higher predicted risk and reasonable PPVs while taking clinical factors that were not included in the model (but which may have improved prediction) into account. However, where the risk is lower, the long time frame of the development of many liver diseases means that GPs can continue to address modifiable liver disease risk factors (eg, alcohol misuse).

The next step is to evaluate ALFI as a complex intervention while taking into account the cost associated with referral of false positives.[38]

GGT should be considered just as important as other LFTs in the prediction of liver disease since only albumin explained more variation in the model. Proper use of ALFI in practice will depend on GGT being a routine part of the LFT panel. Local health economies will have to decide whether the additional cost of an extra test in the panel is worth it in terms of GGT's value in improving the ability to predict those patients who are at high (or low) risk and, subsequently, change referral practice. Before such decisions are made, further research is necessary involving cost-effectiveness analysis.

## CONCLUSIONS

In summary, this study has developed and externally validated the ALFI model for prediction of liver disease diagnosis in patients with no clinically obvious liver disease having their LFTs taken within primary care. From this model, a simple scoring tool was developed to facilitate GP decision-making with regard to retesting or referring their patients. GP decisions regarding referral of patients to secondary care should be based on probability thresholds where benefits outweigh harm.[39] ALFI requires further evaluation as a complex intervention, but it has the potential to save health service costs and prevent unnecessary further investigations and treatments.

**Author affiliations**
[1]Medical Statistics Team, Division of Applied Health Sciences, College of Life Sciences and Medicine, University of Aberdeen, Aberdeen, UK
[2]Dundee Epidemiology and Biostatistics Unit, Division of Population Health Sciences, Medical Research Institute, University of Dundee, Dundee, UK
[3]Department of Community and Family Medicine, University of Toronto, Toronto, Canada
[4]Academic Unit of Primary Care and Population Sciences, Faculty of Medicine, University of Southampton, Southampton, UK
[5]Centre for Hepatology, Division of Medicine and ULCH-UCL NIHR Biomedical Research Centre, University College London, London, UK

[6]Department of Gastroenterology, Nottingham University Hospitals NHS Trust and Biomedical Research Unit, Nottingham, UK
[7]Medical Research Institute, University of Dundee, Dundee, UK

**Acknowledgements** The authors wish to thank Alison Bell from HIC for extracting and pseudonymising all the datasets used. They thank Primary Care Clinical Informatics Unit (PCCIU) at the University of Aberdeen for providing the Practice Team Initiative cohort for external validation of the model.

**Contributors** DJM contributed to the design of the study, conducted the analyses, drafted and finished the paper. JFD, FMS, PR, WMR and SDR contributed to the design and conduct of the study, and the writing of the paper. PTD contributed to and oversaw the design and conduct of the study, as well as contributing to the writing of the paper. He is the guarantor for the paper. All authors read and approved the final manuscript.

**Funding** This work was supported by the UK National Health Service Research & Development Programme Health Technology Assessment Programme (project number 03/38/02) and also by the Backett Weir Russell Career Development Fellowship, University of Aberdeen.

**Competing interests** None.

**Ethics approval** Ethical approval was obtained from the Tayside Committee for Medical Research Ethics in February 2005 (Ref No. 04/S1401/199).

**Provenance and peer review** Not commissioned; externally peer reviewed.

**Data sharing statement** No additional data are available.

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
