## [Reviewer comments · BMJ Open]

Some articles will have been accepted based in part or entirely on reviews undertaken for other BMJ Group journals. These will be reproduced where possible.

ARTICLE DETAILS

TITLE (PROVISIONAL)	Prediction of liver disease in patients whose liver function tests have been checked in primary care: model development and validation using population-based observational cohorts
AUTHORS	McLernon, David; Donnan, Peter; Sullivan, Frank; Roderick, Paul; Rosenberg, William; Ryder, Stephen; Dillon, John

VERSION 1 - REVIEW

REVIEWER	Alan Girling University of Birmingham
REVIEW RETURNED	21-Mar-2014

GENERAL COMMENTS	This paper presents a scoring tool to predict liver disease within two years for patients with LFTs taken in primary care. The model is based on a comprehensive cohort of patients. But the incidence of missing data is very high for two of the LFTs (GGT, 89% missing; Transaminases 23% missing), both of which feature prominently in the model. This is not the authors' fault – more complete data were simply not available – but it could seriously compromise the generalisability of the work to future populations. There is little discussion of the likely cause for the missing data – or more particularly, of the circumstances where the data are present – so that it is impossible to be confident that the conditions for applying imputation techniques are fulfilled. For example, the decision whether or not to order a particular test could itself reflect clinical insight, and focus the sample on patients in which a predictive relationship is most likely to be seen. (Supposing that a particular test is most likely to be ordered in circumstances where it can offer useful discrimination between prognostic groups.) The routine use of the tool in primary care would depend on measuring both GGT and Transaminases in every case, a situation which does not obtain in either the derivation or validation sample. These considerations translate into a serious methodological concern about the validity of the tool. Also, there is a question over whether it would, or could, be used in practice, given that it does seem to ignore clinically relevant information, especially the BMI. This, again, was beyond the authors' control. Detailed points P2 Abstract results. Here, and in other places, the authors make something out of the high NPV for their decision rule. But this is mostly a consequence of the low prevalence (0.5%) in the sample. Any sensible rule would have a NPV of at least 1 minus prevalence i.e. 99.5%, and this figure would be achieved by a null decision rule for which all patients were classified as normal. So the NPV of 99.5% mentioned here is hardly a selling point and does not really support the claim that (on pp 11,16) that NHS costs would be saved. P3 Given that GGT is not routinely tested for, it becomes important
--

	to understand the circumstances where it is done. Its presence in a patient record could itself be clinically informative. P8 No details of the imputation method are given. I.e. at which stages in the model-building was it used, and how were missing values dealt with when constructing the calibration plots? This is more than usually important given the extremely high proportions of missing values, especially for GGT. A more generic point is that imputation methods in general are valid only when Little and Rubin's Missing-at-Random condition is met. This condition cannot be internally verified but it becomes very important when the proportion of missing data is high. Minor P2 Abstract Participants. The phrase "patients having initial abnormal LFTs measured in secondary care" is confusing. It suggests that the study does not include patients with initial LFTs in the normal range, which is not my understanding. P8 (Model assessment) & P11 2nd para. My understanding is that the term "over-fitting" is usually identified with a slope of less than 1 in a calibration plot. P11 3rd line up (and Table A1) Reference to the 75th percentile is a bit confusing in this context. It causes a double-take on first reading (and in the table). Make clear that this refers to the distribution of risk scores in the sample. P12 It might be advisable to show how the clinical scoring tool relates to the final model (e.g. plot of, or at least a correlation between, the two sets of scores.) This is an interesting study, but the issue of generalisability needs to be addressed. It may be that a scoring rule can only apply in the special circumstances where GGT&ASTorALT are routinely ordered, but these circumstances should be described.
--	--

REVIEWER	Gary Collins University of Oxford, UK
REVIEW RETURNED	31-Mar-2014

GENERAL COMMENTS	The authors report on the development and validation of a prediction model for liver disease in a Scottish cohort of patients. The data used to develop the model are 10 years old whilst the validation cohort is more recent. Generally, this is well done and well written, my comments are largely clarifications. Page 7 (Model Derivation) - the author state a parametric regression model was used, could they be more specific in the Methods section, presumably a Weibull model? Page 8. The authors state 'A multiple imputation technique was conducted to impute missing values for the LFTs' - this is rather vague, see Sterne et al BMJ 2009; b2393 for more details on what to report when conducting multiple imputation. Can the authors clarify how the calibration slope was calculated? I would've expected the calibration slope (by definition) to be 1 for the development cohort. Isn't this calculated by fitting the prognostic index (linear predictor) against the outcome? For models built using logistic regression or Cox regression, this would be 1. The use of IDI has started to receive criticisms
---

	Hilden J, Gerds TA (2013) A note on the evaluation of novel biomarkers: do not rely on integrated discrimination improvement and net reclassification index. Stat Med. Hilden J (2014) Commentary: On NRI, IDI, and "Good-Looking" Statistics with Nothing Underneath. Epidemiology 25: 265-267. Kerr KF, McClelland RL, Brown ER, Lumley T (2011) Evaluating the incremental value of new biomarkers with integrated discrimination improvement. Am J Epidemiol 174: 364-374. However, despite these criticisms, ultimately the test of the model is how it performs in an external validation cohort which the authors do. But I just wonder if, in addition to the IDI, the authors should also report the c-index (this is a measure more familiar to readers) I would also suggest examining clinical utility, looking at decision curve analysis (Vickers & Elkin 2006) as a more intuitive way of presenting the model (as a weighted difference between false positives and false negatives) Vickers AJ, Elkin EB (2006) Decision curve analysis: a novel method for evaluating prediction models. Med Decis Making 26: 565-574. To improve uptake of the model, the authors have used the approach by Sullivan et al to create a simplified model. For completeness, what is performance (in the validation cohort) of this simplified model? Presumably there is some (albeit small) deterioration in performance in the simplified model? Figure A3 - I struggle to see the point of a ROC curve in the context of developing a prognostic model. The area underneath the ROC curve, which the authors have reported is summarising these curves. Unless points (i.e. predicted risks as certain points) are presented on the curve, so that sensitivity and specificity can be pulled out, the curves don't really offer anything.
--	--

VERSION 1 – AUTHOR RESPONSE

Reviewer Name Alan Girling

Institution and Country University of Birmingham

Please state any competing interests or state 'None declared': None declared

This paper presents a scoring tool to predict liver disease within two years for patients with LFTs taken in primary care. The model is based on a comprehensive cohort of patients. But the incidence of missing data is very high for two of the LFTs (GGT, 89% missing; Transaminases 23% missing), both of which feature prominently in the model. This is not the authors' fault – more complete data were simply not available – but it could seriously compromise the generalisability of the work to future populations. There is little discussion of the likely cause for the missing data – or more particularly, of the circumstances where the data are present – so that it is impossible to be confident that the conditions for applying imputation techniques are fulfilled. For example, the decision whether or not to order a particular test could itself reflect clinical insight, and focus the sample on patients in which a predictive relationship is most likely to be seen. (Supposing that a particular test is most likely to be ordered in circumstances where it can offer useful discrimination between prognostic groups.) The routine use of the tool in primary care would depend on measuring both GGT and Transaminases in every case, a situation which does not obtain in either the derivation or validation sample. These

considerations translate into a serious methodological concern about the validity of the tool.

RESPONSE: We agree that this is an important point. We have included a table comparing the baseline characteristics of the group of patients with complete data with the group with missing data (see appendix 2). Also included are the characteristics for the groups of patients without GGT, with GGT, without transaminase and with transaminase. As stated in the 'baseline characteristics' section of the results (first sentence on page 13), the table shows that 'the group with complete data were more deprived and contained more alcohol dependent patients than the incomplete data group'. We have also added a further sentence regarding transaminase: 'The group of patients without transaminase measurements contained a higher proportion of females (63.5% versus 36.5%) than that with transaminase measurements (56.2% versus 43.8%) and had a higher median ALP result (86 U/L versus 73 U/L).' In the 'Prediction of liver disease' section of the results we have inserted text describing the results from the model using only complete cases (top of page 14). We have added a new subsection to the discussion which examines missing GGT and transaminase (page 19-20). Although we are not certain, we conclude from discussion with general practitioners that most of the 'missingness' is due to their decision not to order GGT and/or transaminase, as the reviewer correctly states. In Tayside, where the study is set, the laboratories do not routinely include GGT with the other four LFT results unless specifically requested by the primary care physician. The demographics of the patients with complete data (i.e. males, illicit drug users, alcohol dependent people, and patients living in deprived areas) suggested that some primary care physicians may have requested GGT where they suspected substance abuse. Testing bone biochemistry may explain the reason for transaminase not being measured since this group contained a higher proportion of females who are more susceptible to bone disease, such as osteoporosis, and had a higher median alp which is a marker for bone disease. Furthermore, one of the hospitals in Tayside analysed transaminase using a separate analyser for several years throughout the study and did not keep electronic copies which may also explain some of the missing values. Therefore it has been assumed that the missing/untested data depended on variables in the observed data: the missing at random assumption which is required for multiple imputation. Since only albumin explained more variation in the model, we conclude that GGT should be considered just as important as other LFTs in the prediction of liver disease and should be measured more frequently. We agree that the routine use of the tool in primary care would depend on measuring both GGT and Transaminases in every case when the prior probability of disease is similar to the population we studied. We have inserted a new paragraph to the discussion under the section 'Implications to practice and research' with such text and we also state that cost-effectiveness research is required to determine whether GGT is worth being a routine part of the LFT panel (top of page 22).

Also, there is a question over whether it would, or could, be used in practice, given that it does seem to ignore clinically relevant information, especially the BMI. This, again, was beyond the authors' control.

RESPONSE: Some potential predictors of liver disease such as BMI and alcohol intake were not recorded. However, as stated in the 'strength and limitations' section on page 17 we expect the LFTs to act as surrogate markers for this information. For example, alcohol intake can affect GGT and transaminase as can metabolic syndrome (of which increased BMI is a symptom).

Detailed points

P2 Abstract results. Here, and in other places, the authors make something out of the high NPV for their decision rule. But this is mostly a consequence of the low prevalence (0.5%) in the sample. Any sensible rule would have a NPV of at least 1 minus prevalence i.e. 99.5%, and this figure would be achieved by a null decision rule for which all patients were classified as normal. So the NPV of 99.5%

mentioned here is hardly a selling point and does not really support the claim that (on pp 11,16) that NHS costs would be saved.

RESPONSE: We agree that the NPV is a result of the low prevalence of disease and have edited the corresponding text in the discussion to say this (page 20 under 'Implications for practice and research'). Furthermore, although PPV was very low it rose to over 10% for predicted risks over 5% and reached 20-30% for higher risks (over 20%) (see page 15 last paragraph). If the GP was to take clinical factors that weren't measured in the dataset into account (which might have improved prediction and the capacity to benefit from referral) this could help guide the GP to focus on referral on a small minority who at high risk whilst improving the cost effectiveness of the pathway (please see revised 'Implications for practice and research section' on page 19-20). Abstract results have also been edited accordingly. Following reviewer 2's suggestion, we have included a decision curve analysis to determine at what thresholds of predicted probability the model is of value in making decisions regarding referral to secondary care. See Methods (page 11), results (page 16, last sentence of 'Validation' section), discussion (page 21), conclusion and the new figure 1.

P3 Given that GGT is not routinely tested for, it becomes important to understand the circumstances where it is done. Its presence in a patient record could itself be clinically informative.

RESPONSE: Please see our earlier response.

P8 No details of the imputation method are given. I.e. at which stages in the model-building was it used, and how were missing values dealt with when constructing the calibration plots? This is more than usually important given the extremely high proportions of missing values, especially for GGT. A more generic point is that imputation methods in general are valid only when Little and Rubin's Missing-at-Random condition is met. This condition cannot be internally verified but it becomes very important when the proportion of missing data is high.

RESPONSE: We have inserted text in the methods section to explain the multiple imputation procedure that was used (pages 8 (last para) to 9). In the discussion, we discuss the missingness and why we think it is missing at random (MAR not MCAR) (see 'GGT and transaminase' section on page 19-20). Regarding the calibration plots, since we used 30 imputed databases, we decided to avoid applying the final model to each and drawing 30 plots. Instead we decided to take the average imputed LFT value for each patient across the 30 datasets and assess the calibration and discrimination on this dataset. However, we are aware that this is not ideal and since we already have an external cohort with which we have assessed calibration and discrimination we do not think it is even necessary. Therefore we have removed the discrimination and calibration statistics for the final model applied to the averaged development cohort. However, if the reviewer/Editor would prefer us to keep the C-statistic and calibration plot (but not the calibration slope) for this averaged development cohort in the paper then please let us know.

Minor

P2 Abstract Participants. The phrase "patients having initial abnormal LFTs measured in secondary care" is confusing. It suggests that the study does not include patients with initial LFTs in the normal range, which is not my understanding.

RESPONSE: This is indeed an error and we have now removed the word 'abnormal'.

P8 (Model assessment) & P11 2nd para. My understanding is that the term “over-fitting” is usually identified with a slope of less than 1 in a calibration plot.

RESPONSE: We have excluded the calibration slope test from the paper for the development cohort for the reasons given in the response to the last major comment above. Since the calibration slope of the external cohort is less than one this does reflect some over-fitting. However, it was not statistically significant.

P11 3rd line up (and Table A1) Reference to the 75th percentile is a bit confusing in this context. It causes a double-take on first reading (and in the table). Make clear that this refers to the distribution of risk scores in the sample.

RESPONSE: We agree that this was a bit confusing and have edited the sentence to say ‘A cut-off greater than or equal to the 75th percentile of the risk scores (i.e. 0.43% in the derivation cohort and 0.57% in the validation cohort) had sensitivity and specificity of approximately 75% each’ (page 15, last paragraph).

P12 It might be advisable to show how the clinical scoring tool relates to the final model (e.g. plot of, or at least a correlation between, the two sets of scores.)

RESPONSE: The Spearman correlation between the two sets of scores was 0.91 (see appendix 6 for the scatterplot of predicted probability from the final model versus predicted probability from the scoring tool). For high predicted probabilities from the final model, the tool under-estimates the predicted probability. This is because many of these higher probabilities come from patients who have extremely abnormal LFTs. We illustrate this using example cut-offs of very high LFT values represented by the blue circles in the scatterplot (see Appendix 6 for further details). The tool assigns such patients the points based on the highest cut-off in the tool. For example, if a patient has an ALP measure of 825 the tool assigns the same points as someone with an ALP of 181 (i.e. +4 points) since ≥ 180 U/L is the highest cut-off in the tool. It would be inefficient to expand the number of groups for the LFTs to include such extreme values as the score card would be huge. In circumstances where patients have such very large LFT values the GP would automatically know to refer them anyway. Also, the paper which describes how to create this tool (Sullivan et al) recommends reducing the influence of extreme values by using the 99th percentile value of measurements as the largest value when creating the cut-offs. Text relating to this has been added to the results (last paragraph of the results on page 17) and to the ‘Strengths and limitations’ section of the discussion on (bottom of page 18-19).

This is an interesting study, but the issue of generalisability needs to be addressed. It may be that a scoring rule can only apply in the special circumstances where GGT&ASTorALT are routinely ordered, but these circumstances should be described.

RESPONSE: Further to the text above relating to missing GGT and transaminase, we have added a paragraph to the discussion on page 22: ‘GGT should be considered just as important as other LFTs in the prediction of liver disease since only albumin explained more variation in the model. Proper use of ALFI in practice will depend on GGT being a routine part of the LFT panel. Local health economies will have to decide if the additional cost of an extra test in the panel is worth it in terms of GGT’s value in improving the ability to predict those patients who at high (or low) risk and, subsequently, change referral practice. Before such decisions are made further research is necessary involving cost-effectiveness analysis.’

Reviewer Name Gary Collins

Institution and Country University of Oxford, UK

Please state any competing interests or state 'None declared': None declared

The authors report on the development and validation of a prediction model for liver disease in a Scottish cohort of patients. The data used to develop the model are 10 years old whilst the validation cohort is more recent. Generally, this is well done and well written, my comments are largely clarifications.

Page 7 (Model Derivation) - the author state a parametric regression model was used, could they be more specific in the Methods section, presumably a Weibull model?

RESPONSE: Considerably more detail has now been included in the 'Model derivation' section of the methods on page 8. Briefly, a Weibull model was used for the model building. The significant predictors from the Weibull model were then refitted to different parametric model distributions including the generalised gamma, log-logistic, log-normal, and exponential distributions to find the one that fitted best. (as recommended in Dave Collett's textbook – Modelling survival data in medical research). The Akaike's information criterion (AIC) was used to select the optimal model. Covariates which were just outside the significance level for the Weibull model were also added to these other models to check if they became significant. If they did then they were included in that model. The log-normal regression model was found to have the best fit as now mentioned in the first sentence of 'Prediction of a liver disease' section in the results, page 13).

Page 8. The authors state 'A multiple imputation technique was conducted to impute missing values for the LFTs' - this is rather vague, see Sterne et al BMJ 2009; b2393 for more details on what to report when conducting multiple imputation.

RESPONSE: We agree that this is an important point. We have inserted more text in the methods section detailing the multiple imputation procedure that was used (pages 8-9). As answered above for reviewer 1, we have included a table comparing the baseline characteristics of the group of patients with complete data with the group with missing data (see appendix 2). Also included are the characteristics for the groups of patients without GGT, with GGT, without transaminase and with transaminase. As stated in the 'baseline characteristics' section of the results (first sentence on page 13), the table shows that 'the group with complete data were more deprived and contained more alcohol dependent patients than the incomplete data group'. We have also added a further sentence regarding transaminase: 'The group of patients without transaminase measurements contained a higher proportion of females (63.5% versus 36.5%) than that with transaminase measurements (56.2% versus 43.8%) and had a higher median ALP result (86 U/L versus 73 U/L).' In the 'Prediction of liver disease' section of the results we have inserted text describing the results from the model using only complete cases (top of page 14). We have added a new subsection to the discussion which examines missing GGT and transaminase (page 19-20). Although we are not certain, we conclude from discussion with general practitioners that most of the 'missingness' is due to their decision not to order GGT and/or transaminase, as the reviewer correctly states. In Tayside, where the study is set, the laboratories do not routinely include GGT with the other four LFT results unless specifically requested by the primary care physician. The demographics of the patients with complete data (i.e. males, illicit drug users, alcohol dependent people, and patients living in deprived areas) suggested that some primary care physicians may have requested GGT where they suspected substance abuse. Testing bone biochemistry may explain the reason for transaminase not being measured since this group contained a higher proportion of females who are more susceptible to

bone disease, such as osteoporosis, and had a higher median alp which is a marker for bone disease. Furthermore, one of the hospitals in Tayside analysed transaminase using a separate analyser for several years throughout the study and did not keep electronic copies which may also explain some of the missing values. Therefore it has been assumed that the missing/untested data depended on variables in the observed data: the missing at random assumption which is required for multiple imputation. Since only albumin explained more variation in the model, we conclude that GGT should be considered just as important as other LFTs in the prediction of liver disease and should be measured more frequently. We agree that the routine use of the tool in primary care would depend on measuring both GGT and Transaminases in every case when the prior probability of disease is similar to the population we studied. We have inserted a new paragraph to the discussion under the section 'Implications to practice and research' with such text and we also state that cost-effectiveness research is required to determine whether GGT is worth being a routine part of the LFT panel (top of page 22).

Can the authors clarify how the calibration slope was calculated? I would've expected the calibration slope (by definition) to be 1 for the development cohort. Isn't this calculated by fitting the prognostic index (linear predictor) against the outcome? For models built using logistic regression or Cox regression, this would be 1.

RESPONSE: Since we used 30 imputed databases, we decided to avoid applying the final model to each and calculating 30 C-statistics and calibration slopes. Instead we decided to take the average imputed LFT value for each patient across the 30 datasets and assess the calibration and discrimination on this dataset. This would explain why the slope was not equal to one. However, we are aware that this is not ideal and since we already have an external cohort with which we have assessed calibration and discrimination we do not think it is even necessary. Therefore we have removed the discrimination and calibration statistics for the final model applied to the averaged development cohort. However, if the reviewer/Editor would prefer us to keep the C-statistic and calibration plot (but not the calibration slope) for this averaged development cohort in the paper then please let us know. For the validation cohort GGT was missing for 36% of patients and so was imputed using multiple imputation. We decided to simply average the imputed GGT measures across the 30 imputed datasets for each patient who had GGT missing so that validation could be carried out easily in one dataset (see first paragraph of page 11). With 30 imputed datasets we would expect the imputed GGT measures to average to a sensible GGT value.

The use of IDI has started to receive criticisms

Hilden J, Gerds TA (2013) A note on the evaluation of novel biomarkers: do not rely on integrated discrimination improvement and net reclassification index. *Stat Med*.

Hilden J (2014) Commentary: On NRI, IDI, and "Good-Looking" Statistics with Nothing Underneath. *Epidemiology* 25: 265-267.

Kerr KF, McClelland RL, Brown ER, Lumley T (2011) Evaluating the incremental value of new biomarkers with integrated discrimination improvement. *Am J Epidemiol* 174: 364-374.

However, despite these criticisms, ultimately the test of the model is how it performs in an external validation cohort which the authors do. But I just wonder if, in addition to the IDI, the authors should also report the c-index (this is a measure more familiar to readers)

RESPONSE: We have already reported the C-statistic for the final model applied to the external cohort (page 15 2nd para) C=0.78 (95% CI 0.72 to 0.84). Is this what the reviewer is referring to?

I would also suggest examining clinical utility, looking at decision curve analysis (Vickers & Elkin 2006) as a more intuitive way of presenting the model (as a weighted difference between false positives and false negatives)

Vickers AJ, Elkin EB (2006) Decision curve analysis: a novel method for evaluating prediction models. *Med Decis Making* 26: 565-574.

RESPONSE: We liked the reviewers suggestion and conducted a decision curve analysis to determine a range of threshold predicted probabilities of liver disease where the primary care decision to refer a patient to secondary care would be better than assuming all patients are disease free (i.e. not referring anyone) and assuming that all patients have liver disease (i.e. referring everyone). We have added this method to the methods section under the heading 'Decision curve analysis' on page 11. A sentence has been added to the results on page 16 (last sentence of the 'validation' section). A figure displaying the decision analysis curve has been included (Figure 1). The results of the analysis have been discussed on page 21.

To improve uptake of the model, the authors have used the approach by Sullivan et al to create a simplified model. For completeness, what is performance (in the validation cohort) of this simplified model? Presumably there is some (albeit small) deterioration in performance in the simplified model?

RESPONSE: As mentioned above in our response to Reviewer 1, the tool under-predicts where patients have extreme values of LFTs over the 99th percentile. We have calculated the C-statistic for the tool when applied to the external cohort and it is 0.77 (95% CI 0.59 to 0.89) which is only 0.01 lower than the C-statistic for the final model (page 16, last paragraph). When we use the tool's approximation to the linear predictor in a calibration slope test it is not significantly different from one, Beta=0.93 (0.73 to 1.14), p=0.53.

Figure A3 - I struggle to see the point of a ROC curve in the context of developing a prognostic model. The area underneath the ROC curve, which the authors have reported is summarising these curves. Unless points (i.e. predicted risks as certain points) are presented on the curve, so that sensitivity and specificity can be pulled out, the curves don't really offer anything.

RESPONSE: We agree and like the reviewer's idea of plotting risks on the curve. We have edited the ROC curve in Appendix 4 to include some of the predicted risks at certain points.

Elsewhere:

Abstract:

1. Minor edits to make sure that it is clear that the external cohort comprised of 19 general practices across 'the rest of' Scotland (not Tayside where the development cohort came from).
2. Clarity that complications mean 'liver' complications.

Introduction:

1. Grammatical edits in first paragraph (page 4).
- Reference 39 added to the discussion (page 22)